# Faster Approximation Algorithms for $k$-Center via Data Reduction

**Arnold Filtser** [1]  **Shaofeng H.-C. Jiang** [2]  **Yi Li** [3]  **Anurag Murty Naredla** [4]  **Ioannis Psarros** [5]  **Qiaoyuan Yang** [2]
**Qin Zhang** [6]

## Abstract

We study efficient algorithms for the Euclidean $k$-Center problem, focusing on the regime of large $k$. We take the approach of data reduction by considering $\alpha$-coreset, which is a small subset $S$ of the dataset $P$ such that any $\beta$-approximation on $S$ is an $(\alpha + \beta)$-approximation on $P$. We give efficient algorithms to construct coresets whose size is $k \cdot o(n)$, which immediately speeds up existing approximation algorithms. Notably, we obtain a near-linear time $O(1)$-approximation when $k = n^c$ for any $0 < c < 1$. We validate the performance of our coresets on real-world datasets with large $k$, and we observe that the coreset speeds up the well-known Gonzalez algorithm by up to $4$ times, while still achieving similar clustering cost. Technically, one of our coreset results is based on a new efficient construction of consistent hashing with competitive parameters. This general tool may be of independent interest for algorithm design in high dimensional Euclidean spaces.

## 1. Introduction

The $k$-CENTER problem is a fundamental clustering problem that has been extensively studied in various areas, including combinatorial optimization, data science, and machine learning. In $k$-CENTER, the input is a dataset $P \subseteq \mathbb{R}^d$ of $n$ points and a parameter $k$. The goal is to find a set $C \subseteq \mathbb{R}^d$ (called centers) with $|C| = k$ that minimizes the cost function

$$\text{cost}(P, C) := \max_{p \in P} \text{dist}(p, C),$$

where $\text{dist}(p, q) := \|p - q\|_2$ and $\text{dist}(p, C) := \min_{c \in C} \text{dist}(p, c)$. The $k$-CENTER problem presents substantial computational challenges and remains APX-hard even when $d = O(1)$ (Feder & Greene, 1988).

The study of efficient approximation algorithms for $k$-CENTER dates back to the 1980s, with several algorithms providing 2-approximations (Gonzalez, 1985; Hochbaum & Shmoys, 1985; 1986). Among these, the classic algorithm of Gonzalez (Gonzalez, 1985) runs in $O(nkd)$ time. While it achieves good performance for $k = O(1)$, its $O(nk)$ dependence renders it much less efficient for large $k$. In fact, large values of $k$ in $k$-clustering are increasingly relevant in modern applications such as product quantization for nearest neighbor search (Jégou et al., 2011) in vector databases. This has motivated algorithmic studies for the large $k$ regime for $k$-clustering in various computational models (Ene et al., 2011; Bateni et al., 2021; Coy et al., 2023; Czumaj et al., 2024; la Tour & Saulpic, 2024). The record linkage problem, also known as entity resolution or reference reconciliation, has been a subject of study in databases for decades (Koudas et al., 2006; Herzog et al., 2007; Dong & Naumann, 2009). This problem can also be viewed as a $k$-center clustering problem, where $k$ represents the number of ground truth entities and is often very large.

For the large-$k$ regime, it is possible to obtain a subquadratic $\tilde{O}(n^{2-\sqrt{\epsilon}})$ time[1] $(2 + O(\epsilon))$-approximation algorithm[2], and a general ratio-time trade-off as an $O(c)$-approximation in $\tilde{O}(n^{1+1/c^2})$ time (Eppstein et al., 2020)[3]. Yet, it is un-

---

The authors are listed in alphabetical order. [1]Computer Science Department, Bar-Ilan University [2]School of Computer Science, Peking University [3]College of Computing and Data Science, Nanyang Technological University [4]Institut für Informatik, University of Bonn [5]Archimedes, Athena Research Center [6]Luddy School of Informatics, Computing, and Engineering, Indiana University Bloomington. Correspondence to: Arnold Filtser <arnold.filtser@biu.ac.il>, Shaofeng H.-C. Jiang <shaofeng.jiang@pku.edu.cn>, Yi Li <yili@ntu.edu.sg>, Anurag Murty Naredla <anuragmurty@uni-bonn.de>, Ioannis Psarros <ipsarros@athenarc.gr>, Qiaoyuan Yang <qiaoyuanyang@stu.pku.edu.cn>, Qin Zhang <qzhangcs@iu.edu>.

*Proceedings of the 42$^{nd}$ International Conference on Machine Learning*, Vancouver, Canada. PMLR 267, 2025. Copyright 2025 by the author(s).

---

[1]Throughout the paper, the $\tilde{O}$-notation hides the dependence on $\text{poly}(d \log n)$.

[2]This can be obtained by combining an $O(n^{2-\sqrt{\epsilon}})$-time $(1 + O(\epsilon))$-approximate $r$-net construction (Avarikioti et al., 2020) with a standard reduction of $k$-CENTER to net constructions (Hochbaum & Shmoys, 1986).

[3]A similar tradeoff can also be achieved via running an $O(1)$-approximation for sparse graphs (Thorup, 2004) on a Euclidean spanner (Har-Peled et al., 2013).

known if these trade-offs can be improved. The ideal goal is to design an $O(1)$-approximation in near-linear time $\tilde{O}(n)$, which would directly improve the original Gonzalez's runtime by removing an $O(k)$ factor. However, this seems to be challenging with current techniques. Indeed, even the apparently easier task of finding an $O(1)$-approximation to the cost of a *given* center set $C$, without optimizing $C$, in near-linear $\tilde{O}(n)$ time remains unsolved.

In this paper, we present new trade-offs between approximation ratio and running time for $k$-CENTER. Specifically, we focus on optimizing the trade-off in the case of $k = n^c$ ($0 < c < 1$). Our results make a significant step forward towards the ultimate goal of achieving near-linear running time for $O(1)$-approximation in this regime.

## 1.1. Our Results

We take a *data reduction* approach to systematically improve the running time of approximation algorithms for $k$-CENTER. Specifically, we use the notion of an *$\alpha$-coreset* (for $\alpha \geq 1$), defined as a subset $S \subseteq P$ of the dataset $P$ such that any $\beta$-approximate solution to $k$-CENTER on $S$ is an $(\alpha + \beta)$-approximation on the original dataset $P$.

Our main result consists of two coresets with slightly different parameter trade-offs, both of size $k \cdot o(n)$. This essentially reduces the input size from $n$ to $k \cdot o(n)$, speeding up any (existing) approximation algorithm. Notably, we obtain an $O(1)$-approximation in near-linear time for $k = n^c$ ($0 < c < 1$). We summarize the new approximation algorithms in Table 1.

**Coresets.** Our first result, Theorem 1.1, constructs an $O(\alpha)$-coreset of size $O(kn^{1/\alpha^{2/3}})$ in a runtime that is near-linear in $n$ and independent of $\alpha$. Running the existing $\tilde{O}(n^{1+1/\alpha^2})$-time $O(\alpha)$-approximation algorithm (Eppstein et al., 2020) on this coreset, we obtain an $O(\alpha)$-approximation in time $\tilde{O}(n + k^{1+1/\alpha^2} n^{O(1/\alpha^{2/3})})$. This immediately improves the algorithm in (Eppstein et al., 2020). In particular, as long as $k = n^c$ (for any $0 < c < 1$) which can be arbitrarily close to linear $n$, this running time reduces to $\tilde{O}(n)$ by setting $\alpha = \text{poly}((1-c)^{-1})$.

**Theorem 1.1.** *For every $\alpha \geq 1$, there exists an $O(\alpha)$-coreset of size $\tilde{O}(kn^{1/\alpha^{2/3}})$ that can be computed in time $\tilde{O}(n)$ with probability at least $0.99$.*

Our Theorem 1.1 relies on a geometric hashing technique called *consistent hashing* (Czumaj et al., 2022) (see Definition 3.1). Our main technical contribution is to devise a new consistent hashing that offers a competitive parameter trade-off, while still running in $\text{poly}(d)$ time, exponentially improving the previous $\exp(d)$ time construction (Czumaj et al., 2023) (albeit theirs achieves better parameter trade-offs). See Section 3 for a more detailed discussion. This

new hashing result may be useful for algorithm design in high-dimensional Euclidean spaces in general. Finally, we remark that our coreset in Theorem 1.1 may be applied recursively to further reduce the coreset size; see Appendix D for a detailed discussion.

Our second coreset (Theorem 1.2) has size $\tilde{O}(k)$, which is independent of $\alpha$, but has a larger $\tilde{O}(nk^{1/\alpha^2})$ construction time. Running the algorithm in (Eppstein et al., 2020) on this coreset, we obtain an alternative $O(\alpha)$-approximation for $k$-CENTER in $\tilde{O}(nk^{1/\alpha^2})$ time.

**Theorem 1.2.** *For every $\alpha \geq 1$, there exists an $O(\alpha)$-coreset of size $k \cdot \text{polylog}(n)$ that can be computed in time $\tilde{O}(nk^{1/\alpha^2})$ with probability at least $0.99$.*

Previously, it was observed that the point sequence discovered by an (approximate) furthest-neighbor traversal as in Gonzalez's algorithm (Gonzalez, 1985) is an $O(1)$-coreset (Braverman et al., 2021), and one could use an algorithm in (Eppstein et al., 2020) to find such a sequence, which yields an $O(\alpha)$-coreset of size $O(k)$ in time $\tilde{O}(n^{1+1/\alpha^2})$. While this coreset size is competitive, the running time remains super-linear in $k$ for $k = n^c$ ($0 < c < 1$), which is too large for our purpose of near-linear algorithms.

**Experiments.** Our experiments validate the performance of our coresets, with a focus on Theorem 1.1, since Theorem 1.1 leads to near-linear running time for $k$-CENTER when $k = n^c$ ($0 < c < 1$), which is likely to be practical. Our experiments are conducted on four real-world datasets of various sizes and dimensions, and we evaluate the speedup of the well-known Gonzalez's algorithm (Gonzalez, 1985) on our coreset. The experiments show that our coreset provides a consistently better tradeoff between the coreset size and clustering cost than baselines, and runs 2 to 4 times faster than directly running Gonzalez algorithm on the dataset, while still achieving comparable cost values.

## 1.2. Related Work

Our notion of coreset is related to the widely considered *strong* coreset (Agarwal et al., 2004; Har-Peled & Mazumdar, 2004), which is a subset $S \subseteq P$ satisfying that $\text{cost}(S, C) \in (1 \pm \epsilon) \text{cost}(P, C)$ for *all* center sets $C \subseteq \mathbb{R}^d$. The key difference is that ours may not preserve the cost value on $S$ for all $C$, but it does preserve the approximation ratio. Moreover, this stronger notion inherently leads to a prohibitively large coreset size of $\exp(\Omega(d))$, even for $k = 1$.[4] Our notion is sometimes referred to as *weak* coresets in the literature, and similar notions were also considered in (Feldman et al., 2007; Munteanu & Schwiegelshohn, 2018; Huang et al., 2023; Carmel et al., 2025).

---

[4]This lower bound is folklore, but can be easily proved using an $\epsilon$-net on the unit sphere.

Table 1: Summary of approximation algorithms for $k$-CENTER in $\mathbb{R}^d$

| Approx. ratio | Running time | Reference |
|---|---|---|
| 2 | $O(nkd)$ | Gonzalez (1985) |
| $O(\alpha)$ | $\tilde{O}(n^{1+1/\alpha^2})$ | Eppstein et al. (2020) |
| $O(\alpha)$ | $\tilde{O}(n + k^{1+1/\alpha^2} n^{O(1/\alpha^{2/3})})$ | Theorem 1.1 + Eppstein et al. (2020) |
| $O(\alpha)$ | $\tilde{O}(nk^{1/\alpha^2})$ | Theorem 1.2 + Eppstein et al. (2020) |

## 2. Preliminaries

**Notations.** For $m \in \mathbb{N}_{\geq 1}$, denote $[m] := \{1, \ldots, m\}$. For a point set $C \subseteq \mathbb{R}^d$, let $\mathrm{diam}(C)$ denote the diameter of $C$. For $x \in \mathbb{R}^d$ and $r \geq 0$, the ball of radius $r$ centered at $x$ is denoted by $B(x,r) = \{y \in \mathbb{R}^d : \mathrm{dist}(x,y) \leq r\}$, and we write $B_S(x,r) := B(x,r) \cap S$ for $S \subseteq \mathbb{R}^d$. For two sets $A, B \subseteq \mathbb{R}^d$, their Minkowski sum is $A \oplus B := \{x + y : x \in A, y \in B\}$. For a function $f : \mathbb{R}^d \to \mathbb{R}^d$ and a set $S \subseteq \mathbb{R}^d$, we define $f(S) := \{f(x) : x \in S\}$.

**Definition 2.1** (Covering). Given a set $P \subseteq \mathbb{R}^d$ and $\rho \geq 0$, a subset $S \subseteq P$ is called a $\rho$-covering for $P$ if for every $p \in P$, there exists a $q \in S$ such that $\mathrm{dist}(p,q) \leq \rho$.

The following lemma will be useful in both of our coreset constructions. Its proof is deferred to Appendix A.

**Lemma 2.2** (Coarse approximation). *There is an algorithm that, given as input a dataset $P \subset \mathbb{R}^d$ with $|P| = n$ and an integer $k \geq 1$, computes a $\mathrm{poly}(n)$-approximation to the $k$-CENTER cost value with probability at least $1 - 1/\mathrm{poly}(n)$, running in time $O(nd + n\,\mathrm{polylog}(n))$.*

## 3. Efficient Consistent Hashing

The notion of *consistent hashing* was coined in (Czumaj et al., 2023), which partitions $\mathbb{R}^d$ into cells such that each small ball in $\mathbb{R}^d$ intersects only a small number of cells. Partitions with similar properties have also been studied under the notion of *sparse partitions* for general metric spaces (see, e.g., (Jia et al., 2005; Filtser, 2024)). The main differences are that consistent hashing requires the partition to be defined using a (data-oblivious) hash function and emphasizes computational efficiency.

Below we present our formal definition of consistent hashing, which relaxes the definition of Czumaj et al. (2023) by only requiring the number of intersecting cells to be bounded in expectation.

**Definition 3.1.** A $(\Gamma, \Lambda, \ell)$-*consistent hashing* is a distribution over functions $\varphi : \mathbb{R}^d \to \mathbb{R}^d$ such that for every $x \in \mathbb{R}^d$,
- (diameter) $\mathrm{diam}(\varphi^{-1}(x)) \leq \ell$, and
- (consistency) $\mathbb{E}[|\varphi(B(x, \frac{\ell}{\Gamma}))|] \leq \Lambda$.

In this definition, the hash partitions $\mathbb{R}^d$ into buckets of diameter at most $\ell$ (which can be picked arbitrarily), such that any subset in $\mathbb{R}^d$ of diameter at most $\ell/\Gamma$ intersects no more than $\Lambda$ buckets (in expectation). Since consistent hashings are scale invariant in $\mathbb{R}^d$, we omit the parameter $\ell$ in our discussion below. Ours and previous results are summarized in Table 2.

For every parameter $\beta \geq 1$, Filtser (2024) constructed a *deterministic* consistent hashing (namely, the consistency guarantee is worst-case and not in expectation) with parameters $\Gamma = \beta$ and $\Lambda = \tilde{O}(d) \cdot \exp(O(d/\beta))$. However, computing $\varphi(x)$ for a given point $x$ requires both time and space that are exponential in $d$. Nevertheless, Filtser showed that this trade-off between $\Gamma$ and $\Lambda$ is tight up to second order terms regardless of runtime, even when the consistency guarantee is relaxed to expectation only (implicitly). Czumaj et al. (2023) constructed a deterministic consistent hashing with the same parameters, requiring only $\tilde{O}(d^2)$ space, though the function evaluation still takes exponential time in $d$. They also constructed a time- and space-efficient consistent hashing, which can be evaluated in $\mathrm{poly}(d)$ time but with sub-optimal parameters of $\Lambda$ and $\Gamma$.

Our hash function is the first to achieve the bound $\Lambda = \exp(O(d/\beta^c))$ (for some $0 < c \leq 1$) when $\Gamma = \beta$ for every $\beta \geq 1$, while still running in *polynomial* time in $d$. Technically, we construct the hash function using a surprisingly simple randomly-shifted grid, which is widely used in geometric algorithm design.

Previous works also studied laminar consistent hashing (Busch et al., 2012; 2023), which is a sequence of hash functions at different scales, each refining the previous one. We note also that Chen & Zhang (2016) studied a related notion to consistent hashing, but their diameter guarantee was only probabilistic, so it is not directly comparable.

**Lemma 3.2.** *For every $\beta \geq \sqrt{2\pi}$ and $\ell > 0$, there exists a $(\beta, t_\beta, \ell)$-consistent hashing $\varphi : \mathbb{R}^d \to \mathbb{R}^d$ with $t_\beta := \mathrm{poly}(d) \cdot \exp(O(d/\beta^{\frac{2}{3}}))$ which can be computed in $O(d)$ time.*

*Proof.* Since it suffices to define the hash function for an (arbitrary) fixed $\ell$, in this proof we fix $\ell := \sqrt{d}$.

**Construction.** The hash is defined by a randomly-shifted grid. Formally, we first choose a uniformly random vector $v \in [0,1]^d$ and, for each $x \in \mathbb{R}^d$, define $\varphi(x) = \lfloor x +$

Table 2: Summary of results on consistent hashing in Euclidean $\mathbb{R}^d$. The third result is a lower bound.

| $\Gamma$ | $\Lambda$ | Guarantee | Runtime | Space | Reference |
|---|---|---|---|---|---|
| 1 | $\exp(O(d))$ | worst-case | $\exp(d)$ | $\exp(d)$ | Jia et al. (2005) |
| $\beta$ | $\tilde{O}(d) \cdot \exp(O(d/\beta))$ | worst-case | $\exp(d)$ | $\exp(d)$ | Filtser (2024) |
| $\beta$ | $\Lambda > (1 + \frac{1}{2\beta})^d$ | expected (implicit) | N/A | N/A | Filtser (2024) |
| $\beta$ | $\tilde{O}(d) \cdot \exp(O(d/\beta))$ | worst-case | $\exp(d)$ | $\text{poly}(d)$ | Czumaj et al. (2023) |
| $O(d^{1.5})$ | $O(d)$ | worst-case | $\text{poly}(d)$ | $\text{poly}(d)$ | Czumaj et al. (2023) |
| $\beta$ | $\text{poly}(d) \cdot \exp(O(d/\beta^{\frac{2}{3}}))$ | expected | $O(d)$ | $O(d)$ | Lemma 3.2 |

$v \rfloor$. Here, for a vector $z = (z_1, \ldots, z_d) \in \mathbb{R}^d$, we define $\lfloor z \rfloor = (\lfloor z_1 \rfloor, \lfloor z_2 \rfloor, \ldots, \lfloor z_d \rfloor) \in \mathbb{R}^d$, i.e, rounding down $z$ coordinatewise.

**Analysis.** To evaluate $\varphi(x)$, we simply round $x + v$ down coordinatewise to the nearest integer, which takes $O(d)$ time. The diameter property is also straightforward, since $\varphi^{-1}(t) = \times_{i=1}^d [t_i - v_i, t_i - v_i + 1)$ $(t \in \mathbb{Z}^d)$, which is a half-open unit cube and has diameter $\sqrt{d} = \ell$.

It remains to verify that an arbitrary ball of radius $r = \sqrt{d}/\beta$ intersects only $\text{poly}(d)\exp(O(d/\beta^{\frac{2}{3}}))$ grid cells in expectation. Let $x \in \mathbb{R}^d$ be arbitrary and consider the ball $B(x, r)$. Let $\tilde{r} = \lceil r \rceil$. By symmetry, we can assume w.l.o.g. that $x \in [\tilde{r}, \tilde{r} + 1)^d$. Further, for the sake of analysis only, we will slightly change the hash function. Let $\Delta \gg \tilde{r}$ be some fixed large integer to be determined later. Instead of sampling $v \in [0, 1]^d$, we sample $v = (v_1, \ldots, v_d) \in [0, \Delta]^d$ uniformly at random and map each point $y$ to $\lfloor y + v \rfloor$. Note that the number of intersecting grid cells by a ball centered at $(x + v)$ equals the number of intersecting cells by a ball centered at $x + (v_1 \bmod 1, \ldots, v_d \bmod 1)$. Thus, the two hash functions have exactly the same expected consistency.

Since $x \in [\tilde{r}, \tilde{r} + 1)^d$ and $v \in [0, \Delta]^d$, the ball $B(x + v, r)$ can only intersect grid cells in the box $\mathcal{G} = [0, \Delta + 3\tilde{r}]^d$. Fix some grid cell $K = \times_{i=1}^d [t_i, t_i + 1) \subset \mathcal{G}$. Let $X_K$ be an indicator for the event that the ball $B(x + v, r)$ intersects $K$. This happens if and only if the ball $B(v, r)$ intersects the box $\times_{i=1}^d [t_i - x_i, t_i - x_i + 1)$, or, $v$ is contained in the Minkowski sum of the box $\times_{i=1}^d [t_i - x_i, t_i - x_i + 1)$ and the ball $B(\vec{0}, r)$. The following lemma bounds the volume of this Minkowski sum.

**Lemma 3.3** (Aiger et al. (2014), Lemma 3.1)**.** *Let $C = [0, 1]^d$ be the unit cube in $\mathbb{R}^d$ and $0 < r \leq \sqrt{d/2\pi}$ be a parameter. Let $C_r = C \oplus B(\vec{0}, r)$, then*

$$\text{vol}(C_r) \leq \text{poly}(d) \cdot \exp\left(3/2 \cdot (2\pi)^{\frac{1}{3}} d^{\frac{2}{3}} r^{\frac{2}{3}}\right).$$

Applying Lemma 3.3 with our $r = \sqrt{d}/\beta$, we have

$$\text{vol}\left(C_{\sqrt{d}/\beta}\right) \leq \text{poly}(d) \cdot \exp\left(\frac{3}{2}(2\pi)^{\frac{1}{3}} d^{\frac{2}{3}} \cdot (\sqrt{d}/\beta)^{\frac{2}{3}}\right)$$

$$= \text{poly}(d) \cdot \exp\left(\frac{3}{2}(2\pi)^{\frac{1}{3}} d/\beta^{\frac{2}{3}}\right)$$

$$= \text{poly}(d) \cdot \exp\left(O(d/\beta^{\frac{2}{3}})\right).$$

Therefore,

$$\mathbb{E} X_K \overset{(*)}{\leq} \frac{\text{vol}\left([t_i - x_i, t_i - x_i + 1) \oplus B(\vec{0}, r)\right)}{\text{vol}\left([0, \Delta]^d\right)}$$

$$= \frac{\text{vol}(C_r)}{\Delta^d} = \frac{1}{\Delta^d} \cdot \text{poly}(d) \cdot \exp(O(d/\beta^{\frac{2}{3}})).$$

Here $^{(*)}$ is an inequality, rather than equality, because the Minkowski sum $[t_i - x_i, t_i - x_i + 1) \oplus B(\vec{0}, r)$ might not be fully contained in $[0, \Delta]^d$. Only grid cells from $\mathcal{G} = [0, \Delta + 3\tilde{r}]^d$ have a non-zero probability of intersecting $B(x + v, r)$. Since there are only $(\Delta + 3\tilde{r})^d$ such grid cells $K$, by linearity of expectation, the expected number of grid cells intersecting $B(x + v, r)$ is at most

$$(\Delta + 3\tilde{r})^d / \Delta^d \cdot \text{poly}(d) \cdot \exp(O(d/\beta^{\frac{2}{3}}))$$

$$= \text{poly}(d) \cdot \exp(O(d/\beta^{\frac{2}{3}})),$$

where the last equality holds for large enough $\Delta$. This verifies the consistency bound of the consistent hashing and completes the proof of Lemma 3.2. $\qquad\square$

## 4. Proof of Theorem 1.1

We prove Theorem 1.1 in this section (restated below).

**Theorem 1.1.** *For every $\alpha \geq 1$, there exists an $O(\alpha)$-coreset of size $\tilde{O}(kn^{1/\alpha^{2/3}})$ that can be computed in time $\tilde{O}(n)$ with probability at least $0.99$.*

We start by reducing the task of finding coresets to the construction of $\rho$-*coverings* (see Definition 2.1) via a standard fact that any $\alpha$-approximation on an $(\beta \, \text{opt})$-covering is a $(\alpha + \beta)$-approximation to $k$-CENTER (see Lemma 4.1);

hence, it suffices to find a small $(\beta \operatorname{opt})$-covering as a core-set. Indeed, covering is a fundamental notion in geometric optimization. In the context of $k$-CENTER, it can be viewed as a bi-criteria approximation that uses slightly more than $k$ center points.

**Lemma 4.1.** *For a dataset $P \subseteq \mathbb{R}^d$ and integer $k$, consider a $(\beta \operatorname{opt})$-covering $S \subseteq P$ for some $\beta \geq 1$. Then any $\alpha$-approximation on $S$ is an $(\alpha + \beta)$-approximation on $P$ for $k$-CENTER. In other words, $S$ is a $\beta$-coreset.*

*Proof.* For a generic point set $W \subseteq \mathbb{R}^d$ and a point $x \in \mathbb{R}^d$, we define the projection function $\pi_W(x) := \operatorname{argmin}_{y \in W} \operatorname{dist}(x, y)$, which maps $x$ to its nearest neighbor in $W$ (ties are broken arbitrarily). Since $S$ is a $(\beta \operatorname{opt})$-covering, for every $p \in P$ we have $\operatorname{dist}(p, \pi_S(p)) \leq \beta \operatorname{opt}$. Let $\widehat{C}$ be an $\alpha$-approximation to $k$-CENTER on $S$. Then,

$$
\begin{aligned}
\operatorname{cost}(P, \widehat{C}) &= \max_{p \in P} \operatorname{dist}(p, \pi_{\widehat{C}}(p)) \\
&\leq \max_{p \in P} \operatorname{dist}(p, \pi_S(p)) + \operatorname{dist}(\pi_S(p), \pi_{\widehat{C}}(\pi_S(p))) \\
&\leq \max_{p \in P} \operatorname{dist}(p, \pi_S(p)) + \max_{p \in S} \operatorname{dist}(p, \pi_{\widehat{C}}(p)) \\
&\leq \beta \operatorname{opt} + \alpha \operatorname{opt}_S \\
&\leq \beta \operatorname{opt} + \alpha \operatorname{opt},
\end{aligned}
$$

where the last inequality follows from the fact that the optimal $k$-CENTER cost on the subset $S$ cannot be larger than the optimal $k$-CENTER cost on $P$, which is true since we consider the continuous version of the $k$-CENTER problem where centers are chosen from the entire $\mathbb{R}^d$. $\square$

Thanks to Lemma 4.1, it remains to find a small $(\beta \operatorname{opt})$-covering. We give the following construction of covering based on consistent hashing (Definition 3.1). This is the main technical lemma for Theorem 1.1. Its proof is postponed to Section 4.1.

**Lemma 4.2.** *There is an algorithm that takes as input a dataset $P \subseteq \mathbb{R}^d$ with $|P| = n$, $\beta \geq 1$ and integer $k \geq 1$, computes a set $S \subseteq P$ with $|S| \leq k \cdot \operatorname{poly}(d) \exp(d/\beta^{2/3})$ in time $O(nd \log n)$, such that $S$ is an $O(\beta \operatorname{opt})$-covering of $P$ with probability at least $0.991$.*

Lemma 4.2 allows us to compute a covering set whose size is exponential in the dimension (assuming $d \gg \beta$). To mitigate this, we apply the Johnson-Lindenstrauss (JL) transform (Johnson & Lindenstrauss, 1984), using random projections to reduce the dimension of the input point set to $O(\log n)$. The JL Lemma is restated as follows.

**Lemma 4.3** (Johnson-Lindenstrauss Lemma). *Let $P \subseteq \mathbb{R}^d$ be a set of $n$ points and $\epsilon \in (0, \frac{1}{2})$. Then there exists a map $f : P \to \mathbb{R}^{d'}$ for some $d' = O(\epsilon^{-2} \log n)$ such that*

$$(1 - \epsilon) \operatorname{dist}(x, y) \leq \operatorname{dist}(f(x), f(y)) \leq (1 + \epsilon) \operatorname{dist}(x, y)$$

*for all $x, y \in P$. Moreover, the image $f(P)$ can be computed in $O(\varepsilon^{-2} nd \log n)$ time with probability at least $1 - 1/\operatorname{poly}(n)$.*

Now we are ready to conclude the proof of Theorem 1.1.

*Proof of Theorem 1.1.* For a generic point set $W \subseteq \mathbb{R}^d$, let $\operatorname{opt}(W)$ be the optimal $k$-CENTER value on $W$. The algorithm for Theorem 1.1 goes as follows. We first run Lemma 4.3 with some constant $\epsilon = O(1)$ to obtain a mapping $f : \mathbb{R}^d \to \mathbb{R}^{d'}$ where $d' = O(\log n)$. Let $P' := f(P)$ be the dataset in the target space after JL. Then, we apply Lemma 4.2 on $P'$, to obtain an $O(\alpha \operatorname{opt}(P'))$-covering $S' \subseteq P'$ of $P'$. Let $S := f^{-1}(S')$, and this is well-defined since $S'$ is a subset of $P'$ and $f$ is injective on $P$. The algorithm returns $S$ as the covering.

The running time follows immediately from Lemmas 4.2 and 4.3. Next, we verify that $S$ is a desired covering. Conditioning on the success of Lemma 4.3, i.e., for every $x, y \in P$, $\operatorname{dist}(f(x), f(y)) \in (1 \pm \epsilon) \operatorname{dist}(x, y)$, we consider an arbitrary $x \in P$. Then

$$
\begin{aligned}
\operatorname{dist}(x, S) &\leq (1 + \epsilon) \cdot \operatorname{dist}(f(x), S') \\
&\leq (1 + \epsilon) \cdot O(\alpha \operatorname{opt}(P')) \\
&\leq 2(1 + \epsilon)^2 \cdot O(\alpha) \operatorname{opt}(P) \\
&\leq O(\alpha) \operatorname{opt}(P),
\end{aligned}
$$

where the first inequality directly follows from the conditioned event, and the third inequality from the claim that $\operatorname{opt}(P') \leq 2(1 + \epsilon) \operatorname{opt}(P)$, which can be derived from the conditioned event as follows[5]. Consider a 2-approximation $\widehat{C} \subseteq P$ of $k$-CENTER on $P$ (for instance, consider the solution of Gonzalez's algorithm (Gonzalez, 1985)), and the condition implies $\operatorname{opt}(P') \leq \operatorname{cost}(P', f(\widehat{C})) \leq 2(1 + \epsilon) \operatorname{opt}$. Finally, the failure probability follows from a union bound of the failure of Lemmas 4.2 and 4.3. This finishes the proof. $\square$

### 4.1. Proof of Lemma 4.2

**Proof overview.** The covering construction is based on consistent hashing (see Definition 3.1). Consider the $k$ clusters $C_1^*, \ldots, C_k^*$ in an optimal solution, then by definition $\bigcup_i C_i^* = P$ and $\operatorname{diam}(C_i^*) \leq 2 \operatorname{opt}$ for all $C_i^*$. Roughly speaking, the key property of a consistent hashing $\varphi$, is that each $C_i^*$ is mapped to $|\varphi(C_i^*)| \leq \Lambda$ distinct buckets, and that each bucket has diameter $O(\alpha \operatorname{opt})$, where $\Lambda$ is a parameter of the hashing and we have $\Lambda = \operatorname{poly}(d) \cdot \exp(O(d/\beta^{\frac{2}{3}}))$ in our construction (Lemma 3.2). Then, picking an arbitrary point from every non-empty bucket yields an $O(\alpha \operatorname{opt})$-covering of size $k\Lambda$. This hash

---

[5] In fact, one can show $\operatorname{opt}(P') \leq (1 + \epsilon) \operatorname{opt}(P)$, which has also been analyzed in, e.g., (Jiang et al., 2024).

---

**Algorithm 1** Covering based on consistent hashing

---

1: apx $\leftarrow$ a $\gamma$-approximate of $k$-CENTER$(P)$ using Lemma 2.2, where $\gamma = \text{poly}(n)$.
2: $t_\beta \leftarrow \text{poly}(d) \cdot \exp(O(d/\beta^{2/3}))$ (as in Lemma 3.2)
3: **for** $i = 0$ to $\lceil \log \gamma \rceil$ **do**
4: $\quad \tau \leftarrow \frac{\text{apx}}{\gamma} \cdot 2^i$
5: $\quad$ let $\varphi_\tau$ be a $(\beta, t_\beta, \beta\tau)$-hashing sampled using Lemma 3.2
6: $\quad$ for each $z \in \varphi_\tau(P)$, pick an arbitrary *representative* point $\text{rep}(z)$ from the bucket $\varphi_\tau^{-1}(z) \cap P$
7: $\quad$ let $S_\tau \leftarrow \{\text{rep}(z) : z \in \varphi_\tau(P)\}$
8: $\quad$ if $|S_\tau| \leq 200kt_\beta$ then **return** $S_\tau$
9: **end for**

---

$\varphi$ is data-oblivious and we can evaluate $\varphi(x)$ for every $x \in \mathbb{R}^d$ in $O(d)$ time, which leads to a $\tilde{O}(n)$ running time of Lemma 4.2.

**Algorithm.** The algorithm is listed in Algorithm 1. Let opt denote the cost of the optimal solution to the $k$-CENTER on $P$. The algorithm starts by finding a $\text{poly}(n)$-approximation to opt (using Lemma 2.2). It then checks $O(\log n)$ geometrically increasing values of $\tau$, one of which estimates opt up to a factor of 2. For each value $\tau$, we pick a consistent hash $\varphi_\tau$ as in Lemma 3.2, with scale parameter $\beta \cdot \tau$, such that the points of every ball of radius $\frac{\beta \cdot \tau}{\beta} = \tau$ are hashed into only $t_\beta = \text{poly}(d) \cdot \exp(O(d/\beta^{2/3}))$ cells in expectation. For each hash $\varphi_\tau$, the algorithm computes a set $S_\tau$, containing a single representative from every nonempty hash cell. Once $|S_\tau| \leq 10k \cdot t_\beta$, the algorithm halts and returns $S_\tau$.

Consider an estimate $\tau$ such that $\frac{\tau}{2} < \text{opt} \leq \tau$. The points in $P$ are contained in $k$ balls of radius $\text{opt} < \tau$ (around the centers in the optimal solution). Under $\varphi_\tau$, the points within each of these balls are hashed into only $t_\beta$ cells in expectation. This implies that, with high constant probability, $|S_\tau| \leq 200k \cdot t_\beta$, leading the algorithm to halt and return $S_\tau$. We now proceed with a formal proof.

**Lemma 4.4.** *For every $\tau$, the set $S_\tau$ is a $(\beta\tau)$-covering for $P$ (with probability 1).*

*Proof.* Clearly, $S_\tau \subseteq P$. Now, fix some $p \in P$, let $z := \varphi_\tau(p)$. Then $\text{rep}(z) \in S_\tau$, and by Definition 3.1, we have $\text{dist}(p, S_\tau) \leq \text{dist}(p, \text{rep}(z)) \leq \text{diam}(\varphi_\tau^{-1}(z)) \leq \beta\tau$. This verifies the definition of $\beta\tau$-covering. $\square$

**Lemma 4.5.** *For $\tau \geq \text{opt}$, $|S_\tau| \leq 200kt_\beta$ with probability at least 0.995 (over the randomness of $\varphi_\tau$).*

*Proof.* Let $C^* = \{c_1^*, ..., c_k^*\}$ be an optimal solution for $k$-CENTER. Then $P$ can be covered by the $k$ balls of radius

opt around $c_j^*$'s, i.e., $P = \bigcup_{j=1}^k B_P(c_j^*, \text{opt})$. As $\text{opt} \leq \tau$, by linearity of expectation, it holds that

$$\mathbb{E}\left[|S_\tau|\right] = \mathbb{E}\left[|\varphi_\tau(P)|\right] = \mathbb{E}\left[\Big|\varphi_\tau\big(\bigcup_{j=1}^k B_P(c_j^*, \text{opt})\big)\Big|\right]$$

$$\leq \sum_{j=1}^k \mathbb{E}\left[\big|\varphi_\tau(B_P(c_j^*, \text{opt}))\big|\right] \leq k \cdot t_\beta .$$

By Markov's inequality, $\Pr[|S_\tau| > 200kt_\beta] \leq 0.005$. $\square$

*Proof of Lemma 4.2.* We define the two following events:

- $\mathcal{E}_{\text{apx}}$: the event that the $k$-CENTER approximation algorithm in Lemma 2.2 succeeds: $\text{opt} \leq \text{apx} \leq \gamma \cdot \text{opt}$.
- $\mathcal{E}_{\text{hash}}$: the event that for $\tau$ such that $\text{opt} \leq \tau < 2\,\text{opt}$, it holds that $|S_\tau| \leq 200kt_\beta$.

By Lemma 2.2 and Lemma 4.5, with probability at least 0.991, events $\mathcal{E}_{\text{apx}}$ and $\mathcal{E}_{\text{hash}}$ both happen. We now condition on both events and, in the rest of the proof, argue that the algorithm succeeds. Lemma 4.2 will then follow.

**Covering property.** The algorithm iterates over different values of $\tau$, starting at $\tau_0 = \frac{\text{apx}}{\gamma} \leq \text{opt}$, and increase $\tau$ in jumps of 2, with the maximum value being $\tau_{\lceil \log \gamma \rceil} = \frac{\text{apx}}{\gamma} \cdot 2^{\lceil \log \gamma \rceil} \geq \text{apx} \geq \text{opt}$. Let $\tau'$ be the estimate such that $\text{opt} \leq \tau' < 2\,\text{opt}$. If the algorithm will reach the estimate $\tau'$, then as we conditioned on $\mathcal{E}_{\text{hash}}$, the algorithm will halt and return $S_{\tau'}$. Otherwise, the algorithm will halt earlier at some value $\tau \leq \frac{1}{2} \cdot \tau' \leq \text{opt}$. In either case, by Lemma 4.4, the algorithm returns a set $S_\tau$, which is a $(\beta\tau)$-covering for $P$. Note that $\beta\tau \leq 2\beta \cdot \text{opt}$.

**Running time.** The invocation of Lemma 2.2 in Line 1 only takes $O(nd + n\log n)$ time. The for-loop runs at most $O(\log n)$ times, and in each iteration, it takes $O(nd)$ time to evaluate all hash values $\varphi(P)$. In summary, the overall running time of the algorithm is $O(nd\log n)$. This finishes the proof of Lemma 4.2. $\square$

## 5. Constructing Covering via Sampling

We now prove Theorem 1.2 (restated below).

**Theorem 1.2.** *For every $\alpha \geq 1$, there exists an $O(\alpha)$-coreset of size $k \cdot \text{polylog}(n)$ that can be computed in time $\tilde{O}(nk^{1/\alpha^2})$ with probability at least 0.99.*

The proof is similar to that of Theorem 1.1, using a reduction to covering (Lemma 4.1). Hence, the remaining step is to find a suitable covering for Theorem 1.2, which is stated in the following lemma. The lemma relies on an approximate nearest neighbor search (ANN) structure, where given a set of input points $T$ and a query point $q$, the $\alpha$-ANN finds for each a point $x \in T$ such that $\text{dist}(x, q) \leq \alpha \cdot \text{dist}(q, T)$.

**Lemma 5.1.** *Given a set of points $P \subseteq \mathbb{R}^d$, and $k, \beta \geq 1$, there is an algorithm that runs in $\tilde{O}(nk^{1/\beta^2})$ time, and with probability at least $0.991$ returns a set $S \subseteq P$ of $k \cdot \mathrm{polylog}(n)$ points such that $S$ is an $O(\beta\,\mathrm{opt})$-covering of $P$.*

Note that Theorem 1.2 follows directly from Lemmas 4.1 and 5.1. The rest of this section proves Lemma 5.1.

**Proof overview for Lemma 5.1.** Our proof is based on random hitting sets. For the sake of presentation, assume that the $k$ clusters in an optimal solution are of similar size $\Theta(\frac{n}{k})$. Then a uniform sample $S$ of size $O(k \log n)$ would hit all clusters w.h.p. . Furthermore, by the definition of $k$-CENTER, the entire dataset is contained in a $(2\,\mathrm{opt})$-neighborhood of $S$. This $(2\,\mathrm{opt})$-neighborhood of $S$ gives the covering and can be computed using ANN. The general case where the clusters are not balanced can be handled similarly. Specifically, for a random sample $S$ of $O(k \log n)$ points, at least half of the points will, with high probability, be within a distance of $2\,\mathrm{opt}$ from $S$. We can eliminate these points, and repeat this process for $O(\log n)$ rounds to cover all points.

**Algorithm.** The algorithm (Algorithm 2) begins by computing a $\mathrm{poly}(n)$-approximation to $\mathrm{opt}$, denoted by $\mathrm{apx}$. Then it checks $O(\log n)$ geometrically increasing values of $\tau$, one of which estimates $\mathrm{opt}$ up to a factor of 2. For each value of $\tau$, the algorithm attempts to construct a coreset $S_\tau$ such that for every point $x \in P$, $\mathrm{dist}(x, S_\tau) \leq 2\beta\tau$. In more detail, a set $Q$ of uncovered points is maintained. The process consists of $L = O(\log n)$ iterations, where in each iteration, $O(k \log n)$ random (uncovered) points from $Q$ are added to $S_\tau$. ANN is then invoked at Line 6, using the newly sampled points as the input set and the uncovered points in $Q$ as queries, where we use the ANN algorithm of Andoni & Indyk (2006). Every point whose $\beta$-approximate nearest neighbor is within a distance of at most $2\beta\tau$ is subsequently removed from $Q$. If $Q$ becomes empty during the $O(\log n)$ iterations, the algorithm returns $S_\tau$.

For the analysis, consider $\tau \geq \mathrm{opt}$, and $Q \subseteq P$. Since $Q$ can be covered by $k$ balls of radius $\tau$, by an averaging argument, at least half of the points in $Q$ must belong to the balls that each contains at least a $\frac{1}{2k}$ fraction of $Q$. Consequently, each such point will, with constant probability, be within a distance of at most $2\tau$ from the sampled points and will thus be removed from $Q$. It follows that $Q$ is expected to shrink in size by a constant factor in each iteration and, after $O(\log n)$ iterations, becomes empty. The running time is dominated by the executions of ANNs. The next lemma is the main technical guarantee of Algorithm 2 and its proof is postponed to Appendix B.

**Lemma 5.2.** *Suppose that $\tau \geq \mathrm{opt}$. With probability at least $1 - 1/n$, there exists $j \leq L$ such that $Q^{(j)} = \emptyset$ at*

---

**Algorithm 2** Covering based on sampling
1: $\mathrm{apx} \leftarrow$ a $\gamma$-approximate of $k$-CENTER$(P)$ using Lemma 2.2, where $\gamma = \mathrm{poly}(n)$
2: **for** $i \leftarrow 0$ to $\lceil \log \gamma \rceil$ **do**
3: $\quad \tau \leftarrow \frac{\mathrm{apx}}{\gamma} \cdot 2^i$, $\ S_\tau \leftarrow \emptyset$, $\ Q^{(0)} \leftarrow P$
4: $\quad$ **for** $j \leftarrow 1$ to $L = 5\lceil \log n \rceil$ **do**
5: $\quad\quad$ draw $O(k \log n)$ uniform samples $S^{(j)}$ with replacement from $Q^{(j-1)}$
6: $\quad\quad$ for each $x \in Q^{(j-1)}$, compute $\widehat{\mathrm{dist}}(x, S^{(j)})$ such that $\widehat{\mathrm{dist}}(x, S^{(j)}) \leq \beta\,\mathrm{dist}(x, S^{(j)})$ using ANN
7: $\quad\quad$ $R^{(j)} \leftarrow \{x \in Q^{(j-1)} : \widehat{\mathrm{dist}}(x, S^{(j)}) \leq 2\beta\tau\}$
8: $\quad\quad$ $S_\tau \leftarrow S_\tau \cup S^{(j)}$, $\ Q^{(j)} \leftarrow Q^{(j-1)} \setminus R^{(j)}$
9: $\quad\quad$ if $Q^{(j)} = \emptyset$ then **return** $S_\tau$
10: $\quad$ **end for**
11: **end for**

---

Table 3: Specifications of datasets, where $d$ is the original data dimension and $d'$ is the target dimension of the JL transform.

| dataset | size (approx.) | $d$ | $d'$ |
|---|---|---|---|
| Kddcup | 5M | 38 | 30 |
| Covertype | 581K | 55 | 50 |
| Census | 2M | 69 | 60 |
| Fashion-MNIST | 70K | 784 | 100 |

*Line 9.*

*Proof of Lemma 5.1.* During the execution of the algorithm we construct an $\beta$-ANN structure $O(\log^2 n)$ times, each with input size $O(k \log n)$. On each such data structure we perform at most $n$ queries. Specifically, we use the $\beta$-ANN algorithm of Andoni & Indyk (2006), which takes $\tilde{O}(m^{1+1/\beta^2})$ pre-processing time and $\tilde{O}(m^{1/\beta^2})$ query time to compute an $O(\beta)$-approximate NN for each query point in $\mathbb{R}^d$, over an $m$-point input set. This algorithm answers all $O(n)$ queries successfully with $1 - 1/\mathrm{poly}(n)$ probability. Hence, the overall running time is $\tilde{O}(n \cdot k^{1/\beta^2})$ (where we set $m = O(k \log n)$). This also dominates all the other steps.

Let $i \in [0, \lceil \log \gamma \rceil]$ be such that $\frac{\mathrm{apx}}{\gamma} \cdot 2^{i-1} < \mathrm{opt} \leq \frac{\mathrm{apx}}{\gamma} \cdot 2^i$. If the algorithm terminates at iteration $i' < i$, then it has found a set $S_\tau$ which is a $\rho$-covering of $P$ for $\rho = 2\beta \cdot \frac{\mathrm{apx}}{\gamma} \cdot 2^{i'-1} < 2\beta\,\mathrm{opt}$, as claimed. Otherwise, by Lemma 5.2, with probability at least $1 - 1/n$, Algorithm 2 would terminate at the $i$-th iteration and, in this case, $S_\tau$ is a $\rho$-covering of $P$ for $\rho = 2\beta \cdot \frac{\mathrm{apx}}{\gamma} \cdot 2^i \leq 2\beta \cdot 2\,\mathrm{opt} = 4\beta\,\mathrm{opt}$. $\qquad \square$

# 6. Experiments

We implement our coreset from Theorem 1.1 and evaluate its performance by measuring how effective it speeds up the classical algorithm of Gonzalez (1985), which gives a

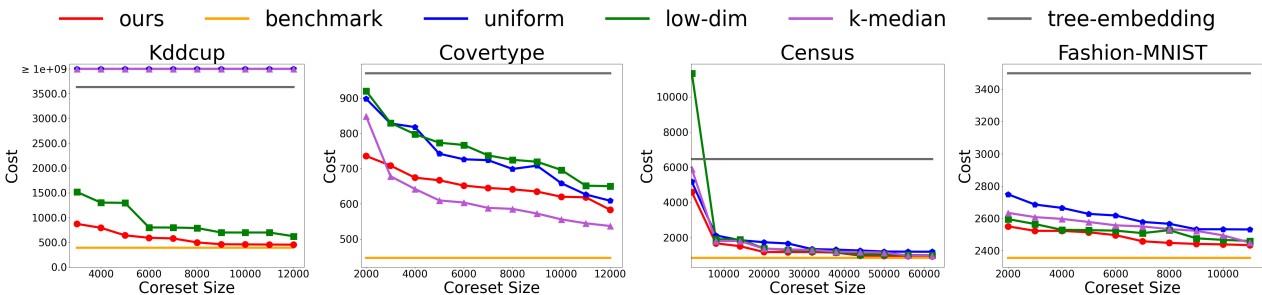

Figure 1: The cost evaluation for all baselines in each dataset.

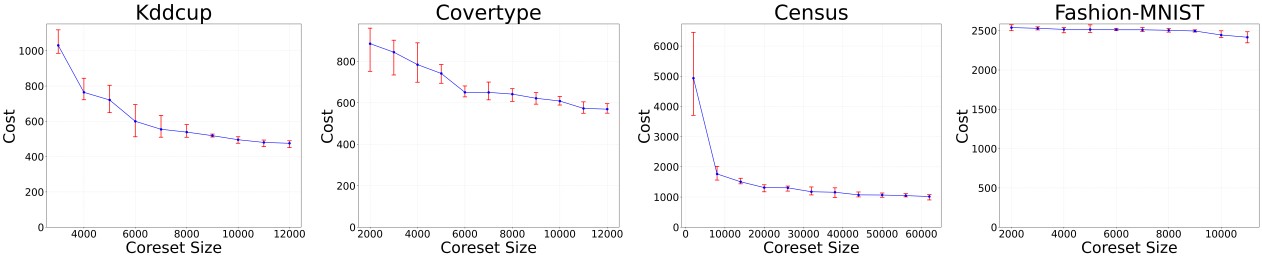

Figure 2: The variance evaluation for our coreset.

2-approximation for $k$-CENTER. We focus on the coreset of Theorem 1.1 due to its near-linear running time and use of simple grid structure, which makes it more practical. Specifically, we conduct experiments with varying coreset sizes and report running time and the corresponding clustering cost when running Gonzalez's algorithm on the coreset. The research code for this experiment is available at a GitHub repository.

**Datasets.** We use four real datasets: Kddcup (Stolfo et al., 1999), Covertype (Blackard, 1998), Census (Meek et al., 2001), and Fashion-MNIST (Xiao et al., 2017). For each dataset, we extract numerical features to construct a vector in $\mathbb{R}^d$ for each record, and we perform a Johnson-Lindenstrauss (JL) transform on each dataset. The detailed dataset specifications as well as the target dimension of the JL transform are summarized in Table 3.

**Implementation details.** The implementation of our coreset mostly follows Algorithm 1 but involves an important modification. Since our experiment is to generate a coreset with a specified size budget $s$ (instead of a pre-defined target error), the error parameter $\beta$, which controls the coreset size, is no longer useful. Consequently, we replace the coreset size upper bound $10kt_\beta$ in Line 8 directly with the budget $s$. We also replace the third parameter $\ell = \beta\tau$ of the hash function in Line 5 with $\tau$. The coresets generated in this manner may not have an exact size of $s$, but it is nonetheless close to $s$.

**Baselines.** We employ five baselines for $k$-CENTER, one is a brute-force benchmark, three are coreset-based, and the other one is based on tree embedding method. The brute-force benchmark, named benchmark, runs Gonzalez's algorithm on the entire dataset. The three baseline coresets are as follows: a) a heuristic coreset based on uniform sampling, called uniform, which samples uniformly a subset of a given size from the dataset; b) a coreset designed for low dimensions, called low-dim, which has a worst-case size of $O(k2^{O(d)})$ (Agarwal et al., 2004); c) a coreset designed for $k$-MEDIAN clustering, called k-median (Chen, 2009; Cohen-Addad et al., 2021). We run a Gonzalez's algorithm on the coresets. The last baseline is based on tree embedding, where we first run a fast tree embedding algorithm via randomly-shifted grid (Indyk, 2004), which has $O(\log^2 n)$ distortion, followed by running Gonzalez's algorithm on the tree embedding, referred to as tree-embedding.

Note that there is no standard way to adapt the low-dim construction to our context; a naïve implementation can lead easily to a coreset size close to $O(k2^{O(d)})$, which is prohibitively large for our datasets. In our experiments, we implement this low-dim baseline in a manner similar to our coreset construction, with the only difference being that we do not use a random shift in the hash function.

**Experiment setup.** For all experiments, we set the number of centers $k \approx \sqrt{n}$, where $n$ denotes the size of the dataset. For coreset baselines, we vary the target coreset size $s$ (ranging from $k$ up to $30k$), compute each baseline coreset with the target size $s$, and evaluate the clustering cost on the full dataset. We report both the running time

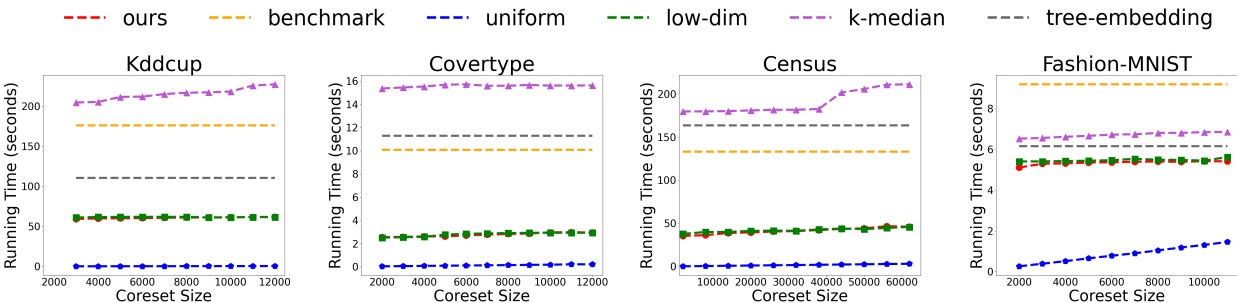

Figure 3: The running time evaluation for all baselines in each dataset.

and the evaluated cost, averaged over 3 independent trials. All algorithms are implemented in C++ and compiled with Apple Clang version 15.0.0 at -O3 optimization level. All the experiments are run on a MacBook Air 15.3 with an Apple M3 chip (8 cores, 2.22 GHz), 16GB RAM, and macOS 14.4.1 (23E224).

**Experiment results.** We depict in Figure 1 the clustering cost of the baselines. More specifically, this also shows a trade-off between the coreset size and the clustering cost for coreset baselines (and for non-coreset baselines `benchmark` and `tree-embedding` we simply plot their cost as a constant function of size). Our coreset uses 5% of the full dataset to achieve a <1.3 times of `benchmark`'s costs on all datasets. Moreover, our algorithm consistently achieves the smallest cost compared with the other `uniform`, `low-dim` and `tree-embedding` baselines, and outperforms `k-median` baseline on most datasets, which confirms the superior performance of our coresets.

We give more discussions on the performance of baselines. The `uniform` baseline performs generally worse than ours and `low-dim` (albeit comparable to `low-dim` in Covertype), which is expected, since naïve uniform sampling may not hit sparse clusters. This is particularly seen in Kddcup dataset, where it has been observed that this dataset has outlier/noisy points (Tavallaee et al., 2009). The performance of `low-dim` is closer to ours, which is an interesting fact, since our implementation helps it to escape the worst-case size of $k \cdot 2^{-O(d)}$ on the tested datasets. As mentioned, the only difference between the implementation of `low-dim` and ours lies in whether or not a random shift is applied in the hash function (Lemma 3.2). Therefore, the performance gain of ours justifies the effectiveness of a random shift even on real-world datasets. The `k-median` baseline achieves lower costs on Covertype. This suggests that the solutions to $k$-MEDIAN and $k$-CENTER are similar on this dataset. The `tree-embedding` baseline performs poorly across all datasets. this is due to the fact that the tree embedding method only provides bounded distortion in expectation, which is insufficient for the $k$-CENTER objective, where the maximum distance is critical.

We also report the variance of our algorithm in Figure 2, which is very small compared to the magnitude of the cost. Finally, we report in Figure 3 the running time, including both coreset construction and the execution of Gonzalez's algorithm, as a function of coreset size. Ours yields a 2x - 4x speedup over `benchmark`, and has generally comparable running time with other baselines albeit achieving a better accuracy as discussed above.

## Acknowledgements

The authors would like to thank the organizers of the "Massive Data Models and Computational Geometry" workshop at the University of Bonn, where the initial discussions leading to this paper took place. Arnold Filtser was supported by the Israel Science Foundation (grant No. 1042/22). Yi Li is supported in part by Singapore Ministry of Education AcRF Tier 2 grant MOE-T2EP20122-0001. Ioannis Psarros has been partially supported by project MIS 5154714 of the National Recovery and Resilience Plan Greece 2.0 funded by the European Union under the NextGenerationEU Program. Qin Zhang is partly supported by NSF CCF-1844234.

## Impact Statement

This paper presents work whose goal is to advance the field of Machine Learning. We do not see any immediate societal consequences of our work.

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

## A. Proof of Lemma 2.2

**Lemma 2.2** (Coarse approximation). *There is an algorithm that, given as input a dataset $P \subset \mathbb{R}^d$ with $|P| = n$ and an integer $k \geq 1$, computes a $\mathrm{poly}(n)$-approximation to the $k$-CENTER cost value with probability at least $1 - 1/\mathrm{poly}(n)$, running in time $O(nd + n\,\mathrm{polylog}(n))$.*

*Proof.* The plan is to first do a random projection to 1D, and show that the pairwise distance is preserved up to $\mathrm{poly}(n)$ factor. This step takes $O(nd)$ time. Then we can apply an off-the-shelf near-linear $n \cdot \mathrm{polylog}(n)$ time algorithm for $k$-CENTER on 1D (Megiddo & Tamir, 1983).

Specifically, let $v \in \mathbb{R}^d$ be a random vector with entries sampled independently from a standard Gaussian distribution, i.e., $v \sim \mathcal{N}(0, I_d)$. For every point in $P$, compute its inner product with $v$, resulting in $P'$, i.e., $P' = \{\langle p, v\rangle : \forall p \in P\}$. $P'$ can be interpreted as the projection of $P$ onto one-dimension. It remains to prove the following distortion bound.

**Claim.** $\Pr\left[\forall x, y \in P, \ \frac{\mathrm{dist}(x,y)}{\mathrm{poly}(n)} \leq |\langle v, x\rangle - \langle v, y\rangle| \leq \mathrm{dist}(x,y) \cdot \mathrm{poly}(n)\right] \geq 1 - 1/\mathrm{poly}(n)$.

*Proof of Claim.* The claim trivially holds if $x = y$, and hence we assume $x \neq y$. Let $u := (\langle v, x\rangle - \langle v, y\rangle)/\mathrm{dist}(x,y)$. Then by a standard property of Gaussian, $u \sim \mathcal{N}(0, 1)$. On the one hand, by Markov's inequality, we immediately obtain $\Pr[|u| \geq \mathrm{poly}(n)] \leq 1/\mathrm{poly}(n)$ since $\mathbb{E}[|u|] = \Theta(1)$. On the other hand, since the density function of $N(0, 1)$ is upper bounded by $1/\sqrt{2\pi}$, we know that $\Pr[|u| \leq 1/\mathrm{poly}(n)] \leq 1/\mathrm{poly}(n)$ by integrating the density function from $-1/\mathrm{poly}(n)$ to $1/\mathrm{poly}(n)$. This finishes the proof. $\qquad\square$

$\qquad\qquad\qquad\qquad\qquad\qquad\qquad\qquad\qquad\qquad\qquad\qquad\qquad\qquad\qquad\qquad\qquad\qquad\qquad\qquad\square$

## B. Proof of Lemma 5.2

**Lemma 5.2.** *Suppose that $\tau \geq \mathrm{opt}$. With probability at least $1 - 1/n$, there exists $j \leq L$ such that $Q^{(j)} = \emptyset$ at Line 9.*

*Proof.* Fix some iteration $j$ in the inner for-loop (Line 4). We claim that

$$\Pr\left[|Q^{(j)}| \leq \frac{1}{2} \cdot |Q^{(j-1)}|\right] \geq \frac{1}{2} \,. \tag{1}$$

That is, in any given iteration, the size of $Q^{(j)}$ decreases by a factor of at least 2 with probability at least $\frac{1}{2}$. Over the $L$ iterations of the for-loop, if this size reduction occurs in at least $\log n$ iterations, then $Q^{(j')}$ will be the empty set for some $j' \leq L$. Assuming that inequality (1) holds, the probability that the size reduction occurs less than $\log n$ times is negligibly small:

$$\Pr\left[Q^{(L)} \neq \emptyset\right]$$
$$\leq \sum_{q=0}^{\lceil\log n\rceil} \binom{L}{q} \cdot \frac{1}{2^L} \overset{(*)}{\leq} \frac{1}{2^L}\left(\frac{L \cdot e}{\lceil\log n\rceil}\right)^{\lceil\log n\rceil}$$
$$= \frac{1}{32^{\lceil\log n\rceil}} \cdot (5e)^{\lceil\log n\rceil} < \frac{1}{2^{\lceil\log n\rceil}} \leq \frac{1}{n} \,,$$

where $(*)$ follows from the bound $\sum_{i=0}^{k} \binom{n}{i} \leq (\frac{en}{k})^k$ (see, e.g., Exercise 0.0.5 in Vershynin (2018)).

It remains to prove inequality (1). Let $\{c_1^*, \ldots, c_k^*\}$ be an optimal solution for $k$-CENTER on $P$, thus $Q^{(j-1)} \subseteq P \subseteq \bigcup_{q=1}^{k} B_P(c_q^*, \mathrm{opt})$. Let $C_1, C_2, \ldots, C_k$ be a partition of $Q^{(j-1)}$ such that $C_q \subseteq B_P(c_q^*, \mathrm{opt})$ for all $q \in [k]$. In particular, $\mathrm{diam}(C_q) \leq 2\,\mathrm{opt}$ for each $C_q$. We say that a cluster $C_q$ is *large* if $|C_q| \geq \frac{|Q^{(j-1)}|}{2k}$, and *small* otherwise. Denote by $\mathcal{C}$ the set of large clusters. Note that the number of points in small clusters is at most $k \cdot \frac{|Q^{(j-1)}|}{2k} = \frac{|Q^{(j-1)}|}{2}$. Let $\Psi$ be the event that the set $S^{(j)}$ of samples contains at least one point from each large cluster. By a union bound, the probability that $\Psi$ does not happen is

$$\Pr\left[\overline{\Psi}\right] = \Pr\left[\exists\,\text{large cluster } C_q \text{ s.t. } C_q \cap S^{(j)} = \emptyset\right]$$

$$\leq \sum_{C_q \in \mathcal{C}} \Pr\left[C_q \cap S^{(j)} = \emptyset\right]$$

$$= \sum_{C_q \in \mathcal{C}} \left(1 - \frac{|C_q|}{|Q^{(j-1)}|}\right)^{O(k \cdot \log n)}$$

$$\leq k \cdot \left(1 - \frac{1}{2k}\right)^{O(k \cdot \log n)} = \frac{k}{n^{O(1)}} \leq \frac{1}{n} .$$

Next, condition on $\Psi$. Then $S^{(j)}$ contains a point from $C_q$ for every large cluster $C_q$. Since each cluster has diameter at most $2\,\mathrm{opt}$ and more than half of the points belong to large clusters, it follows that there are at least $\frac{|Q^{(j-1)}|}{2}$ points in $Q^{(j-1)}$, each of which is within a distance of at most $2\,\mathrm{opt} \leq 2\tau$ from some point in $S^{(j)}$. Our ANN will return with high probability, for each $x \in Q^{(j)}$, an estimate $\widehat{\mathrm{dist}}(x, S^{(j)})$ such that $\widehat{\mathrm{dist}}(x, S^{(j)}) \leq \beta \, \mathrm{dist}(x, S^{(j)})$. It follows that $\widehat{\mathrm{dist}}(x, S^{(j)}) \leq 2\beta\tau$ for all the large cluster points. Consequently, all these points will be included in $R^{(j)}$ and thus $|Q^{(j)}| = |Q^{(j-1)} \setminus R^{(j)}| \leq \frac{|Q^{(j-1)}|}{2}$, with probability at least $\frac{1}{2}$, as claimed. $\qquad\square$

## C. Composability and Reducibility of Covering

Note that our $\alpha$-coresets in both Theorem 1.1 and Theorem 1.2 for a point set $P$ are specifically $(\alpha \, \mathrm{opt})$-covering for $P$. We show that such covering is both composable and reducible, in the following claims. Combining both claims, our coreset may be plugged in a merge-and-reduce framework (Har-Peled & Mazumdar, 2004), which has been used to obtain streaming (Har-Peled & Mazumdar, 2004), dynamic (Henzinger & Kale, 2020) and distributed (Balcan et al., 2013) algorithms for clustering, to imply algorithms for $k$-CENTER in the mentioned settings.

However, we note that the claimed composability and reducibility may not hold directly from the definition of our coreset (which is more general than covering).

**Claim C.1.** *For a generic point set $W \subseteq \mathbb{R}^d$, let $\mathrm{opt}(W)$ be the optimal $k$-CENTER cost on $W$. Consider two datasets $A, B \subseteq \mathbb{R}^d$, and suppose $S_A, S_B$ are $(\alpha \, \mathrm{opt}(A))$-covering for $A$ and $(\alpha \, \mathrm{opt}(B))$-covering for $B$, respectively. Then, $S_{AB} := S_A \cup S_B$ is an $(\alpha \, \mathrm{opt}(A \cup B))$-covering for $A \cup B$.*

*Proof.* We verify the definition. Consider any point $x \in A \cup B$, then

$$\mathrm{dist}(x, S_{AB}) = \mathrm{dist}(x, S_A \cup S_B) = \min\{\mathrm{dist}(x, S_A), \mathrm{dist}(x, S_B)\} \leq \min\{\alpha \, \mathrm{opt}(A), \alpha \, \mathrm{opt}(B)\} \leq \alpha \, \mathrm{opt}(A \cup B).$$

This finishes the proof. $\qquad\square$

We also give the following claim on the reducibility of covering.

**Claim C.2.** *Consider a dataset $P \subseteq \mathbb{R}^d$, and suppose $S$ is an $(\alpha \, \mathrm{opt}(P))$-covering on $P$. Then any $(\beta \, \mathrm{opt}(S))$-covering on $S$ is an $(\alpha + \beta) \, \mathrm{opt}(P)$-covering on $P$.*

*Proof.* We verify the definition. Consider an arbitrary $(\beta \, \mathrm{opt}(S))$-covering $S'$ on $S$. For a generic set $W \subseteq \mathbb{R}^d$ and $y \in \mathbb{R}^d$, let $W(y)$ denote the nearest neighbor of $y$ in $W$. Then for every $x \in P$, we have

$$\mathrm{dist}(x, S') = \mathrm{dist}(x, S'(x)) \leq \mathrm{dist}(x, S(x)) + \mathrm{dist}(S(x), S'(x)) \leq \alpha \, \mathrm{opt}(P) + \beta \, \mathrm{opt}(S) \leq (\alpha + \beta) \, \mathrm{opt}(P).$$

This finishes the proof. $\qquad\square$

## D. Recursive Application of Theorem 1.1

In the $k = n^{1-\epsilon}$ regime which we focus on, a recursive application of Theorem 1.1 will lead to an improved coreset size, as well as a respective improvement in the runtime for the $k$-CENTER algorithm. The optimal number of recursion iterations depends on $k$. In what follows we provide a more detailed explanation.

Denote by $n_j$ the coreset size after $j$ applications of our algorithm in Theorem 1.1. The first coreset is of size roughly $n_1 \sim k n^{\alpha^{-2/3}}$, running it again will get us an $O(\alpha)$-coreset of size $n_2 \sim k \cdot \left(k n^{\alpha^{-\frac{2}{3}}}\right)^{\alpha^{-\frac{2}{3}}} = k^{1+\alpha^{-\frac{2}{3}}} \cdot n^{\alpha^{-\frac{4}{3}}}$, and this is

an improvement when $k \leq n^{1-\alpha^{-\frac{2}{3}}}$. In general, for any fixed $j$, one can apply the algorithm recursively $j$ times and get an $O(j \cdot \alpha)$ coreset of asymptotic size $n_j \sim k^{\sum_{q=0}^{j-1} \left(\alpha^{-\frac{2}{3}}\right)^q} \cdot n^{\left(\alpha^{-\frac{2}{3}}\right)^j} = k^{\frac{1-(\alpha^{-2/3})^j}{1-\alpha^{-2/3}}} n^{(\alpha^{-2/3})^j}$. This is beneficial as long as $k \leq n_j^{1-\alpha^{-\frac{2}{3}}}$. However, one should note that our notation hides polylogarithmic factors that will accumulate. Thus one should use the recursion only a constant number of times.

Applying (Eppstein et al., 2020) on top of the resulting coreset after $j$ iterations will lead to an $O(j \cdot \alpha)$ approximation algorithm for $k$-CENTER with running time bounded by $\tilde{O}\left(n + n_j^{1+\alpha^{-2}}\right) = \tilde{O}(n) + \tilde{O}\left(k^{\frac{1-(\alpha^{-2/3})^j}{1-\alpha^{-2/3}}} n^{(\alpha^{-2/3})^j}\right)^{1+\alpha^{-2}} \leq \tilde{O}\left(n + k^{\frac{1+\alpha^{-2}}{1-\alpha^{-2/3}}} n^{\alpha^{-\frac{2j}{3}} \cdot (1+\alpha^{-2})}\right)$, for any fixed $j$.

