# OpenReview forum: "Faster Approximation Algorithms for k-Center via Data Reduction"
_ICML.cc/2025/Conference — ICML 2025 poster_

### Official Review · Reviewer_MqS4 · 2025-02-15

**Overall Recommendation:** 4

**Summary:**

This paper presents fast algorithms for approximate $k$-center in Euclidean spaces. In general metric spaces, it is not possible to be faster than $n\cdot k$ for any bounded approximation. A folklore result in Euclidean spaces yields a $O(n\cdot(k + d)\cdot\varepsilon^{-2}\log n)$ for a $2+\varepsilon$ approximation. Following a standard JL mapping to low dimensions, the authors give a running time of $O(n + k^{1+1/\alpha^2}n^{\alpha^{-2/3}})$ (which I and I imagine also the authors consider the main result) and an alternative algorithm running in time $O(nk^{1/\alpha^2})$.

The main result is based on an efficient implementation of consistent hashing: a bucketing technique that allows for fast approximate covering of the data set.

**Claims And Evidence:**

The results are nice and as far as I can tell correct. I would be a bit more excited if the the authors had managed to get a clean $\tilde{O}(n+k^{1+1/\alpha^2})$ or even a $O(n+\text{poly}(k))$ bound, as long as the approximation factor is non-trivial. Applying their ideas recursively would maybe be a step in that direction (see further below).

The experiments could be a bit better (see further below).

I ultimately decided on an accept. The result is strong, even it could be improved and well within the scope of ICML.

**Essential References Not Discussed:**

As a alternative for deriving the $O(n^{1+1/\alpha^2})$ bound, one could also first construct a spanner for high dimensional Euclidean spaces by Sariel Har-Peled, Piotr Indyk, Sidiropoulos (SODA'13) and then run an algorithm for k-center in sparse graphs by Thorup (SICOMP'04). Since these results were published earlier than the Eppstein, Har-Peled, Sidiropoulos paper, they should at least be mentioned.

**Experimental Designs Or Analyses:**

-

**Methods And Evaluation Criteria:**

-

**Other Comments Or Suggestions:**

-

**Other Strengths And Weaknesses:**

-

**Questions For Authors:**

The experiments could be better. The authors mainly compare between coreset-based approaches, but this leaves out other option. While the uniform sampling can serve as a sanity check for "is the data set trivial to cluster" and should be included, I am not sure what the point of the low-dimensional coreset construction is, which is designed for small approximation factors.

A more interesting comparison would have been for using a tree embedding, which has $poly(\log n)$ distortion, followed by a linear time algorithm for k-center in trees. These algorithms are almost certain to be equally fast or perhaps even faster than what the authors are presenting, especially given the large values of $k$ which is the regime of interest here. If a tree-embedding based approach is similarly accurate, it would be good to know. I would recommend the paper even if the tree-embedding results were empirically better, as they will yield provably worse theoretical bounds.

One other question that immediately came to my mind is why the main result cannot be applied recursively. While the approximation will deteriorate, the dependency on $n$ could be reduced even more. I see no clear reason why that should not work and it would be nice if the authors either included it (with a discussion on how the parameters would change) or explained why it wouldn't work.

**Relation To Broader Scientific Literature:**

Trading accuracy for running time has been an important topic, especially recently. In particular linear time or nearly linear time algorithms tend to be the most useful results for implementation. This paper gives adds to this research.

While I wouldn't consider k-center the most interesting objective in this line of research and it is not clear whether ideas for k-center extend to other problems, it is neverthess important.

**Theoretical Claims:**

-

---

> ### Author Rebuttal · Authors · 2025-03-31
>
> We thank the reviewer for insightful comments. We address the main concerns as follows.
>
> > As a alternative for deriving the  bound, one could also first construct a spanner for high dimensional Euclidean spaces by Sariel Har-Peled, Piotr Indyk, Sidiropoulos (SODA'13) and then run an algorithm for k-center in sparse graphs by Thorup (SICOMP'04). Since these results were published earlier than the Eppstein, Har-Peled, Sidiropoulos paper, they should at least be mentioned.
>
> Thanks for suggesting these references. We will add these in the next version.
>
> > The experiments could be better. The authors mainly compare between coreset-based approaches, but this leaves out other option. While the uniform sampling can serve as a sanity check for "is the data set trivial to cluster" and should be included, I am not sure what the point of the low-dimensional coreset construction is, which is designed for small approximation factors.
>
> Although conceptually the low dimension baseline is not very relevant to our regime, we find the construction algorithm is in fact very similar to our high dimensional coreset. In particular, in an optimized variant/implementation of the low dimensional baseline, it is the same with our high-dimensional algorithm except that the random shift is skipped (and is replaced with a fixed grid). It is thus interesting to compare with such a baseline to demonstrate the usefulness of the random shift.
>
> > A more interesting comparison would have been for using a tree embedding ... which has $\mathrm{polylog}(n)$ approximation ...
>
> First, we note that while tree embedding has $O(\log n)$ distortion, it has bounded distortion only in the expected sense, and this is too weak for $k$-center where we care about the **max** distance. Therefore, we think that tree embedding cannot yield a finite ratio for $k$-center, at least not $\mathrm{polylog}(n)$.
>
> Nonetheless, we still try to add tree embedding as a new baseline, and we rerun the experiment on the Fashion-MNIST baseline. The initial result is listed here: [Cost evaluation with new baselines -- Fashion-MNIST](https://github.com/r8mtL0pks/ICML25-paper8893-rebuttal/blob/main/All%20Baselines%20(Fashion-MNIST).png). Indeed, it shows that tree embedding performs poorly on this dataset.
>
> > One other question that immediately came to my mind is why the main result cannot be applied recursively. While the approximation will deteriorate, the dependency on $n$ could be reduced even more. I see no clear reason why that should not work and it would be nice if the authors either included it (with a discussion on how the parameters would change) or explained why it wouldn't work.
>
> We thank the reviewer for the interesting suggestion. The short answer is that in some parameter regimes (and in particular in the  $k = n^{1 - \epsilon}$ regime which we focus on), a recursive application will lead to an improved coreset size, and a respective improvement in the runtime for the $k$-center algorithm. The optimal number of recursion iterations depends on $k$. In what follows we provide a more detailed explanation.
>
> Denote by $n_j$ the coreset size after $j$ applications of our algorithm.
> The first coreset is of size roughly $n_1\sim k n^{\alpha^{-2/3}}$, running it again will get us an $O(\alpha)$-coreset of size $n_{2}\sim k\cdot\left(kn^{\alpha^{-\frac{2}{3}}}\right)^{\alpha^{-\frac{2}{3}}}=k^{1+\alpha^{-\frac{2}{3}}}\cdot n^{\alpha^{-\frac{4}{3}}}$, and this is an improvement when $k\le n^{1-\alpha^{-\frac{2}{3}}}$.
> In general, for any fixed $j$, one can apply the algorithm recursively $j$ times and get an $O(j\cdot\alpha)$ coreset of asymptotic  size $n_{j}\sim k^{\sum_{q=0}^{j-1}\left(\alpha^{-\frac{2}{3}}\right)^{q}}\cdot n^{\left(\alpha^{-\frac{2}{3}}\right)^{j}}=k^{\frac{1-(\alpha^{-2/3})^{j}}{1-\alpha^{-2/3}}}n^{(\alpha^{-2/3})^{j}}$. This is beneficial as long as  $k\le n_j^{1-\alpha^{-\frac{2}{3}}}$. However, one should note that our notation hides polylogarithmic factors that will accumulate. Thus one should use the recursion only a constant number of times.
>
> Applying [EHS20] on top of the resulting coreset after $j$ iterations will lead to an $O(j\cdot\alpha)$ approximation algorithm for $k$-center with running time bounded by $\tilde{O}\left(n+n_{j}^{1+\alpha^{-2}}\right)=\tilde{O}(n)+\tilde{O}\left(k^{\frac{1-(\alpha^{-2/3})^{j}}{1-\alpha^{-2/3}}}n^{(\alpha^{-2/3})^{j}}\right)^{1+\alpha^{-2}}\le\tilde{O}\left(n+k^{\frac{1+\alpha^{-2}}{1-\alpha^{-2/3}}}n^{\alpha^{-\frac{2j}{3}}\cdot(1+\alpha^{-2})}\right)$, for any fixed $j$.
>
> We will do a more detailed calculation and add it to the next version.

---

### Official Review · Reviewer_WZSx · 2025-03-07

**Overall Recommendation:** 4

**Summary:**

This paper studies fast algorithms for the k-Center problem, focusing on the regime of large k. It presents a novel approach based on the corsets that achieves better running time.

**Claims And Evidence:**

The claims are supported by the evidence.

**Essential References Not Discussed:**

No.

**Experimental Designs Or Analyses:**

Yes.

**Methods And Evaluation Criteria:**

Yes.

**Other Comments Or Suggestions:**

No

**Other Strengths And Weaknesses:**

Strengths:
- The problem considered is important, and the improvements in running time are significant.
- The result is clearly stated, and the paper is well-written.
- The regime of k considered is interesting.
- The novelty of the work is on par with the expectations of this conference.

Weaknesses:
- The algorithm achieves a constant factor approximation but not 2.
- In some cases, the performance is close to or even worse than the baselines.

**Questions For Authors:**

I think the paper is in a good shape and I do not have any questions.

**Relation To Broader Scientific Literature:**

This work is important and impactful in the field of k-center.

**Theoretical Claims:**

Yes, the paper is sound.

---

> ### Author Rebuttal · Authors · 2025-03-31
>
> We thank the reviewer for insightful comments. We address the main concerns as follows.
>
> > The algorithm achieves a constant factor approximation but not 2.
>
> We agree that achieving a factor of $2$ in near-linear time when $k = n^{1 - \epsilon}$ is an ideal goal and is certainly an interesting open question. Although our work does not directly resolve it, we still offer relevant techniques for attacking the problem, which has potential to lead to small factor approximation (e.g., a ratio of $4$).

---

> > ### Comment · Reviewer_WZSx · 2025-04-02
> >
> > I checked the comments from other reviewers and decided to keep the score.

---

### Official Review · Reviewer_RhVR · 2025-03-14

**Overall Recommendation:** 3

**Summary:**

The paper proposes speeding up existing algorithms for the $k$- center problem by using coresets. An $\alpha$-coreset for $k$- center problem is a subset of data such that if you have a $\beta$ approximation for the $k$- center problem on the subset then you have an $(\alpha + \beta) $ approximation for $k$-center on the full data. The authors build coresets using the notion of consistent hashing. They experimentally demonstrate the efficiency of their coresets both in terms of speed up as well as accuracy preservation on real data sets.

**Claims And Evidence:**

The paper is written clearly and is not very difficult to follow. I find the proofs and the ideas convincing enough.

**Essential References Not Discussed:**

Not to the best of my knowledge

**Experimental Designs Or Analyses:**

See methods and evaluation criteria.

**Methods And Evaluation Criteria:**

The datasets used in the experiments are standard in the literature. In terms of methods, in coreset literature, people typically obtain centers from the coreset and then report the loss on full data using those centers and compare it with the cost on full data when the problem is directly solved for the full data. Are the same quantities shown in the graphs in Figure 1?  It is not clear to me.

The authors could have used more sampling techniques as baselines for example they could have used coresets for other clustering problems like $k$-means, $k$- median and used them as baselines also along with uniform sampling.

**Other Comments Or Suggestions:**

1) The paper relies heavily on the idea of consistent hashing which is pretty recent. It would be good to give some more intuitive explanation of this ideas. For e.g. what $\Gamma $ and $\Lambda$ mean intuitively.

**Other Strengths And Weaknesses:**

See responses to other sections

**Questions For Authors:**

1) If I understood correctly, in the definition of consistent hashing a smaller $\Lambda$ is better? Am I correct? If yes, how is the $\Lambda$ you obtain better than the existing one, since the parameter $c$ is less than 1.

2) How does your algorithm perform/ compare with others in case of small and moderate $k$- values?

3) While I understand hiding the dependence on $\log n$ in the $\tilde {O}$ notation, I am not sure about the dependence on $d$. How will your algorithm perform compared with the Gonzalez and Eppstein algorithm in terms of $d$, especially as we may encounter problems where both $k$ and $d$ are large and using the JL Lemma will have impact on accuracy.

**Relation To Broader Scientific Literature:**

$k$- center is a very well-studied problem and the Gonzalez 2 -approximation algorithm is also well known. The paper combines ideas form literature on high dimensional data science for e.g., the JL -Lemma, covering, hashing etc. to come up with algorithms that speed up the Gonzalez algorithm without harming the accuracy too much. Specifically, the idea is to obtain an almost linear time algorithm for a constant factor approximation for the $k$- center problem for large values of $k$.

**Theoretical Claims:**

Most of the proofs are available in the main body of the paper. They are not too difficult to follow for most parts. I checked them to the best of my ability, and they appear correct.

---

> ### Author Rebuttal · Authors · 2025-03-31
>
> We thank the reviewer for insightful comments. We address the main concerns as follows.
>
> > In terms of methods, in coreset literature, people typically obtain centers from the coreset and then report the loss on full data using those centers and compare it with the cost on full data when the problem is directly solved for the full data. Are the same quantities shown in the graphs in Figure 1? It is not clear to me.
>
> We do compare the cost in the full data. We will clarify this in the next version.
>
> > The authors could have used more sampling techniques as baselines for example they could have used coresets for other clustering problems like $k$-means, $k$-median and used them as baselines also along with uniform sampling.
>
> We added $k$-median coreset as a baseline, based on the ring sampling method (see e.g. [1][2]) which is widely used in various recent works. We obtain an initial result for the cost evaluation on the Fashion-MNIST dataset, and our result can be found in the following: [Cost evaluation with new baselines -- Fashion-MNIST](https://github.com/r8mtL0pks/ICML25-paper8893-rebuttal/blob/main/All%20Baselines%20(Fashion-MNIST).png). We will complete the experiment on the other datasets in the next version.
>
> In short, this $k$-median coreset baseline performs similarly to the uniform sampling baseline, and is worse than our algorithm.
>
> [1] Ke Chen. On Coresets for k-Median and k-Means Clustering in Metric and Euclidean Spaces and Their Applications. SIAM Journal on Computing, 2009.
>
> [2] Vincent Cohen-Addad, David Saulpic, Chris Schwiegelshohn.
> A new coreset framework for clustering. STOC 2021.
>
>
>
> > The paper relies heavily on the idea of consistent hashing which is pretty recent. It would be good to give some more intuitive explanation of this ideas. For e.g. what $\Gamma$  and $\Lambda$  mean intuitively.
>
> The consistent hashing is a space partition, where each part is a bucket. Ideally, we wish each part has a bounded diameter $\ell$ (which can be picked arbitrarily), and that any subset $S$ that has diameter $O(\ell)$ is completely contained in one bucket -- this ensures consistency in hashing subsets of $O(\ell)$ diameter.
>
> Compared with the ideal case, what we actually obtain is relaxed in two aspects: a) the guarantee of consistency is relaxed to intersecting $\Lambda$ buckets instead of only one, and b) the subset $S$ needs to be small enough, whose diameter must at most $\ell / \Gamma$, instead of $O(\ell)$.
>
> We will add this explanation in the next version.
>
> > If I understood correctly, in the definition of consistent hashing a smaller $\Lambda$ is better? Am I correct? If yes, how is the $\Lambda$  you obtain better than the existing one, since the parameter $c$ is less than 1.
>
> Indeed, our $\Lambda$ is not better than the state of the art. However, the state of the art that achieves the best $\Lambda$ has a $2^d$ running time which is too costly. Instead, the focus of this paper is to find an efficient construction that runs in $\mathrm{poly}(d)$ time, at the cost of a worse yet still useful parameter.
>
> > How does your algorithm perform/ compare with others in case of small and moderate $k$ values?
>
> We already provided a brief discussion of this in the introduction. Basically, for moderate $k = n^{1 - \epsilon}$ (where $0 < \epsilon <1$ is arbitrary), our algorithm can achieve $\mathrm{poly}(1 / \epsilon)$-approximation in $\tilde{O}(nd)$ time. This is better than the other works we mention, which either runs in $O(nk) = O(n^{2 - \epsilon})$ time, or in $n^{1 + \mathrm{poly}(\epsilon)}$ time, but not near-linear in $n$ as we do.
>
> > While I understand hiding the dependence on $\log n$ in the $\tilde O$ notation, I am not sure about the dependence on $d$. How will your algorithm perform compared with the Gonzalez and Eppstein algorithm in terms of $d$, especially as we may encounter problems where both $k$ and $d$ are large and using the JL Lemma will have impact on accuracy.
>
> Indeed, we did not attempt to optimize the dependence of $d$ in Theorem 1. The dominating dependence of $d$ comes from Lemma 3.3 (which is a mathematical fact about Minkowski sum), and in other places our algorithm runs (nearly) linear in $d$. Therefore, the dependence of $d$ in the worst-case bound is worse than Gonzalez.
>
> However, in practice one may not need to follow the worst-case upper bound of Lemma 3.3. In particular, in our implementation (in the experiments), for any given coreset size, our dependence of $d$ is near-linear.

---

### Official Review · Reviewer_R2wq · 2025-03-17

**Overall Recommendation:** 3

**Summary:**

The paper "Faster Approximation Algorithms for k-Center via Data Reduction" presents a study on efficient algorithms for solving the Euclidean k-Center problem, focusing particularly on large values of k. The main contribution of the paper is the development of approximation algorithms using a data reduction approach. The authors introduce the concept of α-coresets, which are small subsets of the dataset that maintain the approximation characteristics of the full dataset. Specifically, an α-coreset ensures that a β-approximation on the subset provides an (α+β)-approximation on the original dataset.

The authors propose two coresets with sizes of  k·o(n), significantly reducing the problem size and thus speeding up the existing approximation algorithms. Their approach leads to a near-linear time O(1)-approximation when k = n^c for 0 < c < 1. Through extensive experiments, they demonstrate that these coresets can accelerate the well-known Gonzalez algorithm by 2-4 times, while achieving comparable clustering costs.
One key technical contribution is a new efficient construction of consistent hashing, which is used to build these coresets. This method provides competitive parameters while running in polynomial time, which is a substantial improvement over previous exponential-time constructions.

**Claims And Evidence:**

The claims made in the paper "Faster Approximation Algorithms for k-Center via Data Reduction" are generally well-supported by clear and convincing evidence. The authors provide several forms of evidence to back their claims:
1. Mathematical Proofs: The core idea of constructing α-coresets is supported by rigorous mathematical proofs, particularly in Theorems 1.1 and 1.2, where the existence of efficient coresets and their size bounds are formally established. These theorems are a foundational aspect of the paper's approach, and the authors provide detailed proofs for their correctness.
2. Experimental Validation:The experimental results provide strong empirical support for the claims. The authors demonstrate that their coreset approach significantly speeds up Gonzalez's algorithm for large values of kk achieving a 2-4x speedup while maintaining comparable clustering costs.
3. Theoretical Comparisons: The authors also compare their approach with prior work, citing improvements over existing approximation algorithms and the reduction of time complexity. The comparison of their coreset's performance against other methods, such as uniform sampling, further strengthens their claims.

However, there are a few claims that could be considered more difficult to evaluate fully without additional clarification or further experimentation:
1. General Applicability of Coresets: While the paper demonstrates the utility of their coreset approach for large k. the generalizability of the approach to all clustering problems (or problems with significantly different characteristics) could be better explored. The paper focuses on the Euclidean k-Center problem, so it’s unclear how easily these methods could extend to non-Euclidean metrics or more complex clustering structures.
2. Scalability to Extremely Large Datasets:The experiments show speedups for moderately large datasets. However, it would be valuable to have further evidence on how the approach scales with extremely large datasets (e.g., millions of points and very high-dimensional spaces). Since real-world applications often deal with such data sizes, this would provide more insight into the practicality of the approach.
3. Failure Probability: The paper mentions a failure probability for certain parts of the algorithm, but a more detailed discussion on how this failure probability affects the overall performance, particularly in practical settings, would be useful.

**Essential References Not Discussed:**

The paper provides a thorough review of the relevant literature, and I did not find any essential references that were missing. The key contributions are well-situated within the context of prior work.

**Experimental Designs Or Analyses:**

The experimental design and analyses in the paper are generally sound and valid. However, more experimental results should be added (too few now).
1. The paper uses four real-world datasets—Kddcup, Covertype, Census, and Fashion-MNIST. These datasets are widely used in machine learning and clustering tasks, making them appropriate benchmarks for evaluating the performance of the proposed method. The datasets are diverse in terms of size and dimensionality, which allows the authors to test their method on both small and large datasets.
2. The paper evaluates the proposed method using two primary metrics: “Clustering cost” (i.e., the approximation quality) by running the Gonzalez algorithm on the coreset; “Running time” for constructing the coreset and executing the clustering algorithm. These are standard and appropriate evaluation criteria for clustering problems. The clustering cost is essential for assessing the quality of the approximation, and the running time is crucial for understanding the computational efficiency of the approach, especially when the goal is to improve performance for large-scale datasets.

**Methods And Evaluation Criteria:**

Yes, the proposed methods and evaluation criteria in the paper make sense for the problem at hand, and they are well-suited to evaluate the effectiveness of the algorithm.

**Other Comments Or Suggestions:**

I have no additional comments or suggestions.

**Other Strengths And Weaknesses:**

Strengths:
1. The paper’s novel combination of coreset construction and consistent hashing for improving k-Center algorithms is a major strength. The idea of constructing efficient coresets for large-scale clustering problems is not entirely new, but the specific combination of techniques (such as consistent hashing with competitive parameters and data reduction through α-coresets) brings a new level of efficiency to the problem.The use of geometric hashing for constructing the coreset is another unique aspect of the paper. The new consistent hashing method introduced has competitive parameters and runs in polynomial time, improving upon previous methods that were either slower or had less desirable trade-offs.
2. The paper is well-structured and clearly written. The key ideas are presented logically, with a clear distinction between the theoretical foundations (coreset construction, consistent hashing) and their practical applications (speedup of the Gonzalez algorithm).The experiments and results are presented in a clear, understandable manner, with sufficient explanation of the trade-offs between coreset size, clustering cost, and running time. The inclusion of real-world datasets adds practical context and strengthens the overall contribution.
3. The extensive experimental validation provides strong evidence for the effectiveness of the proposed approach. The authors demonstrate a 2-4x speedup in clustering time compared to the Gonzalez algorithm while maintaining similar clustering costs. These empirical results support the theoretical claims and show that the algorithm works well in practice.

Weaknesses:
1. While the paper demonstrates significant speedups for moderately large datasets, more exploration of edge cases would be beneficial. For example, how does the algorithm perform on datasets with extreme characteristics, such as sparse or imbalanced data? These scenarios are common in many real-world applications, and understanding how the algorithm handles them could further demonstrate its robustness.
2. The paper doesn’t fully address how well the method handles noisy data or missing values, which are often present in practical datasets. Given that many real-world applications involve imperfect data, providing results for these scenarios would make the paper’s findings even more impactful.
3. The paper mentions a failure probability for some of the algorithmic steps, but a more detailed discussion of how this impacts overall performance (especially in real-world applications) would be useful. Including error bounds or confidence intervals for the reported performance would help quantify the robustness of the approach. The statistical significance of the experimental results is not addressed. While the authors run multiple trials for their experiments, providing error bars or statistical tests would improve the reliability of the findings, especially when comparing performance across datasets or baselines.

**Questions For Authors:**

In recent yerars, few works concentrate on this area. How can we agree with the need to implement research on this area? Or, in the other words, can you add more ecessity and significance about your work and applications?

**Relation To Broader Scientific Literature:**

The authors advance the state of the art in k-Center clustering by introducing a novel approach that combines data reduction (via α-coresets and consistent hashing) with efficient approximation algorithms. This approach addresses key challenges in scaling k-Center algorithms to large datasets and high-dimensional spaces, making it a significant contribution to the fields of approximation algorithms, clustering, and high-dimensional data analysis. The connection to earlier work on coresets, approximation algorithms, and dimensionality reduction is clear, and the paper builds on these ideas to propose new, more efficient techniques for large-scale clustering.

**Theoretical Claims:**

I find the theoretical claims in the paper to be reasonable and well-supported by mathematical proofs and established algorithmic techniques. The core ideas (coreset construction, efficient approximation algorithms, and consistent hashing) appear logically sound, and the proofs align with known results in the field of approximation algorithms for clustering.

---

> ### Author Rebuttal · Authors · 2025-03-31
>
> We thank the reviewer for insightful comments. We address the main concerns as follows.
>
> >  General Applicability of Coresets: ... The paper focuses on the Euclidean k-Center problem, so it’s unclear how easily these methods could extend to non-Euclidean metrics or more complex clustering structures.
>
> Our techniques may extend to $\ell_p$ metrics, for example $\ell_1$. Indeed, the main Euclidean structure that we utilize is the consistent hashing, and such hashing with competitive bounds for $\ell_p$ spaces were known to exist, see e.g. [1] (albeit one may need to devise efficient constructions of the hashing, as in our paper).
>
> [1] Arnold Filtser. Scattering and Sparse Partitions, and Their Applications. ACM Transactions on Algorithms, 2024.
>
> > Scalability to Extremely Large Datasets: ... However, it would be valuable to have further evidence on how the approach scales with extremely large datasets (e.g., millions of points and very high-dimensional spaces). Since real-world applications often deal with such data sizes, this would provide more insight into the practicality of the approach.
>
> We expect that our algorithm would be competitive even in this extreme case.
>
> On the one hand, the high dimensionality $d$ is a somewhat less important challenge in our case, since one may apply Johnson-Lindenstrauss Transform to make $d = O(\log n)$, so we do not expect super-high $d$ would have a significant impact to our algorithm.
>
> On the other hand, for the case of $n$ being millions of points, we already demonstrated the performance of our algorithm in this case, in our kddcup dataset which has $n = 5M$, and $d = 38$.
>
> > Failure Probability: The paper mentions a failure probability for certain parts of the algorithm, but a more detailed discussion on how this failure probability affects the overall performance, particularly in practical settings, would be useful. ... Including error bounds or confidence intervals for the reported performance would help quantify the robustness of the approach. The statistical significance of the experimental results is not addressed. While the authors run multiple trials for their experiments, providing error bars or statistical tests would improve the reliability of the findings, especially when comparing performance across datasets or baselines.
>
> First of all, we already take the failure probability into account in our experiments, and we report the cost etc. after we run the algorithms multiple times and take the average.
>
> We also added new experiments to show the variance of the cost of our algorithm. Please check the following for the results of each dataset:
>
> [Census](https://github.com/r8mtL0pks/ICML25-paper8893-rebuttal/blob/main/Cost%20vs.%20Coreset%20Size%20with%20Variance%20(Census).png)
>
> [Covertype](https://github.com/r8mtL0pks/ICML25-paper8893-rebuttal/blob/main/Cost%20vs.%20Coreset%20Size%20with%20Variance%20(Covertype).png)
>
> [Fashion-MNIST](https://github.com/r8mtL0pks/ICML25-paper8893-rebuttal/blob/main/Cost%20vs.%20Coreset%20Size%20with%20Variance%20(Fashion-MNIST).png)
>
> [KDDCup](https://github.com/r8mtL0pks/ICML25-paper8893-rebuttal/blob/main/Cost%20vs.%20Coreset%20Size%20with%20Variance%20(Kddcup).png)
>
> In short, the variance is very small compared with the magnitude of the cost.
>
> > how does the algorithm perform on datasets with extreme characteristics, such as sparse or imbalanced data? These scenarios are common in many real-world applications, and understanding how the algorithm handles them could further demonstrate its robustness.
> ... The paper doesn’t fully address how well the method handles noisy data or missing values, which are often present in practical datasets. Given that many real-world applications involve imperfect data, providing results for these scenarios would make the paper’s findings even more impactful.
>
> We actually find imbalanced distribution/outliers in our KDDCup dataset. This also a way to explain why the uniform baseline performs so bad in this dataset (since it misses the extreme point which may affect the $k$-center objective significantly). Nonetheless, our algorithm performs consistently well even in this dataset.
>
> The missing value seems to be an independent challenge, since it usually requires modeling the missing value to make the problem well-defined. Hence, this is indeed out of the scope of our current algorithm, but it is nonethelss interesting to explore how to model missing values for $k$-center.

---

> > ### Comment · Reviewer_R2wq · 2025-04-04
> >
> > The authors have provided a thoughtful and detailed rebuttal, addressing several key concerns. They clarified the applicability of their approach to high-dimensional data and large-scale datasets, providing additional experimental validation and discussing their algorithm's stability. However, some concerns remain. While they acknowledged challenges related to imbalanced and noisy data, their discussion lacks concrete experimental evidence on these aspects. Overall, the rebuttal improves the clarity and confidence in the work but does not fully resolve all concerns. I keep my original score.

---

### Decision · Program_Chairs · 2025-05-01

**Decision:**

Accept (poster)

**Comment:**

The paper proposes a new algorithm for Euclidean k-center with constant approximation and faster runtime in the large k regime. In this problem, we are given n points in the Euclidean space and the goal is to select k centers in the ambient space so that the maximum distance between any point in the dataset and the nearest center is minimized. The result is achieved via new consistent hashing constructions. All reviewers appreciate the importance of the problem and the theoretical improvement. The theoretical approximation factor is a large constant factor but in experiments, the new algorithm seems competitive with the classical method while improving the runtime by up to a factor 4. The reviewers had some concerns regarding the experiments but this aspect has been improved in the rebuttals.